# Pollinator Proboscis Length Plays a Key Role in Floral Integration of Honeysuckle Flowers (*Lonicera* spp.)

**DOI:** 10.3390/plants12081629

**Published:** 2023-04-12

**Authors:** Gan-Ju Xiang, Amparo Lázaro, Xiao-Kang Dai, Jing Xia, Chun-Feng Yang

**Affiliations:** 1College of Life Science, Yangtze University, Jingzhou 434025, China; 2Global Change Research Group, Mediterranean Institute of Advanced Studies (IMEDEA; UIB-CSIC), 07190 Esporles, Balearic Islands, Spain; 3Department of Biology, Ecology Area, University of the Balearic Islands, 07190 Palma, Balearic Islands, Spain; 4College of Horticulture and Forestry Sciences, Huazhong Agricultural University, Wuhan 430070, China; 5Hubei Provincial Key Laboratory for Protection and Application of Special Plant Germplasm in Wuling Area of China, College of Life Sciences, South-Central Minzu University, Wuhan 430074, China; 6CAS Key Laboratory of Aquatic Botany and Watershed Ecology, Wuhan Botanical Garden, Chinese Academy of Sciences, Wuhan 430074, China; 7Center of Conservation Biology, Core Botanical Gardens, Chinese Academy of Sciences, Wuhan 430074, China

**Keywords:** floral integration, *Lonicera*, phylogenetic structural equation models, pollinator proboscis

## Abstract

Pollinator-mediated selection is supposed to influence floral integration. However, the potential pathway through which pollinators drive floral integration needs further investigations. We propose that pollinator proboscis length may play a key role in the evolution of floral integration. We first assessed the divergence of floral traits in 11 *Lonicera* species. Further, we detected the influence of pollinator proboscis length and eight floral traits on floral integration. We then used phylogenetic structural equation models (PSEMs) to illustrate the pathway through which pollinators drive the divergence of floral integration. Results of PCA indicated that species significantly differed in floral traits. Floral integration increased along with corolla tube length, stigma height, lip length, and the main pollinators’ proboscis length. PSEMs revealed a potential pathway by which pollinator proboscis length directly selected on corolla tube length and stigma height, while lip length co-varied with stigma height. Compared to species with short corolla tubes, long-tube flowers may experience more intense pollinator-mediated selection due to more specialized pollination systems and thus reduce variation in the floral traits. Along elongation of corolla tube and stigma height, the covariation of other relevant traits might help to maintain pollination success. The direct and indirect pollinator-mediation selection collectively enhances floral integration.

## 1. Introduction

Phenotypic integration refers to the coordinated variation of morphological traits within functional modules and results from the simultaneous occurrence of historical, physiological, developmental, and adaptive processes [1,2,3,4]. When phenotypic integration is the consequence of natural selection, acting on the functioning of those modules, it is called functional integration [2,5,6]. Disentangling the factors that influence phenotypic integration can enhance our understanding of the evolution of organs with convoluted modules.

Flowers are an ideal model system for studying the evolution of phenotypic integration. On the one hand, modules within a flower may be highly correlated due to genetic or developmental correlations [7,8]. On the other hand, flowers are functional structures, consisting of different modular units that play roles on different functions, such as pollinator attractiveness [9,10], pollinator accessibility [11], and pollen transfer (pollination efficiency) [12]. Pollinator-mediated selection may help to enhance floral integration [3,13,14]. However, natural selection usually acts on individual floral traits or on a set of traits within flowers, rather than integrating the whole floral structure [12,14,15,16]. For example, in *Ipomoea*, floral traits with pollen transfer function could display greater integration than those involved in pollinator attraction [12]. Pollination is known to contribute to the integration of floral traits. However, the pathway by which pollinators mediate floral integration is still unclear.

Pollinator proboscis has been suggested to play a key role in the evolution of corolla tube (or spur) length [17,18,19,20]. Elongation of the corolla tube may help in filtering out pollinators, so that less efficient ones cannot access to floral nectar. Some studies have indicated that pollinator proboscis length might influence floral integration. For instance, Gómez et al. [4] found that, in Brassicaceae, plants with long corolla tubes were mostly visited by pollinators with long probosces (e.g., hoverflies, large bees) and showed higher floral integration than plants with short corolla tubes. Compared to flowers with short corolla tubes, long-tubed flowers may be visited by a narrower spectrum of pollinators and, thus, may experience stronger selection [21] and display less variable flower sizes [22]. This indicates that pollinators’ selection on corolla tube may play driving key roles in integrating floral traits. However, understanding how such a selection pressure enhances floral integration requires a macroevolutionary framework that incorporates several species.

We assessed the relationship between pollinators, corolla tube length, and floral integration by using 11 *Lonicera* species (Caprifoliaceae). The genus is composed of approximately 180 species, and it is widely distributed in temperate and subtropical areas, with several species having ranges that extend into tropical areas of India, Malaysia, and the Philippines [23,24]. It is traditionally divided into two subgenera, *Lonicera* (Linn.) Rehd. and *Caprifolium* L. [25]. The subgenus *Lonicera* is distinguished by two-flowered cymes and free leaves, whereas the subgenus *Caprifolium* has three-flowered cymes in whorls, and it has perfoliating leaves, subtending the inflorescences [26]. Flowers of the genus have copious colorful corollas that are white, yellow, reddish, or purple-red, sometimes changing color after anthesis. Each flower produces five stamens and one pistil with a capitate stigma [23,25,27]. Moreover, within this genus, there is a significant interspecific difference in shape and size. For example, corolla tubes often range from slightly to deeply gibbous (7 mm–90 mm) on the ventral side toward the base, and they are rarely spurred (Flora of China, http://www.efloras.org/, accessed on 23 January 2020). Petals can be zygomorphic or actinomorphic, and the length of corolla tube varies highly across species [25,28]. *Lonicera* species have a wide range of pollinators, such as bees, moths, and hummingbirds [28,29,30]. Therefore, honeysuckle flowers provide an ideal system to study floral evolution and its relationship with pollinators.

In this study, we conducted field observations and measurements in 11 *Lonicera* species to assess whether pollinators influenced floral integration, and, if so, to illustrate the potential pathway by which pollinators might integrate floral traits. The 11 *Lonicera* study species differed considerably in floral traits (Appendix A). These species are protandrous, self-incompatible, and exclusively dependent on animal pollinators, which provide both pollen and nectar for pollinators, for sexual reproduction [28,30]. Our particular objectives were: (1) to evaluate whether pollinator proboscis was related to corolla tube length and influenced floral integration, and in the case that it did, (2) to figure out the pathway by which pollinator proboscis enhanced floral integration.

## 2. Results

### 2.1. The Divergence of Floral Traits among 11 Lonicera Species

Results of ANOVAs showed that the floral traits (corolla tube length, throat diameter, anther height, stigma height upper corolla lip length/width, and lower corolla lip length/width) differed significantly among species (*F*_10, 299_ ≥ 85.717, *p* < 0.001; see Table 1 for average values of the floral traits for the 11 species). Results from Tukey tests indicated that *L. japonica* and *L. tragophylla* had higher average values for most of the floral traits (except for throat diameter and upper corolla lip width) than the other species (Table 1). The result of PCA showed that the 11 *Lonicera* species were separated into three distinct regions in the morphospace (Figure 1). The first principal component (PC1) explained 93.69% of variance among plant species in floral traits. Based on PC1, *L. japonica* and *L. tragophylla* were separated from the other species. The most important traits to separate species along PC1 were corolla tube length and stigma height (Figure 1); they were greater in *L. japonica* and *L. tragophylla* than in the other species (Table 1). Moreover, PCA analysis indicated that corolla tube length, stigma height, and anthers’ height were the floral traits that most contributed to variation among the 11 *Lonicera* species (Figure 1).

### 2.2. The Relationship between Floral Integration and Floral Traits and Pollinators’ Proboscis Length

The magnitude of floral integration varied over 7.47-fold range; the value for *L. webbiana* (INT = 9.346) was the minimum, while *L. tragophylla* exhibited the maximum value (INT = 33.724; Table 1). Results of PGLS (phylogenetic generalized least squares) revealed that floral integration was positively related to corolla tube length (PGLS: model = OU, b = 0.417 ± 0.166, t = 2.513, *p* = 0.0332; Figure 2a), stigma height (PGLS: model = PL, b = 0.255 ± 0.092, t = 2.768, *p* = 0.0218; Figure 2b), upper lip length (PGLS: model = PL, b = 0.775 ± 0.196, t = 3.96, *p* = 0.0033; Figure 2c), and lower lip length (PGLS: model = PL, b = 0.693 ± 0.187, t = 3.711, *p* = 0.0048; Figure 2d). Floral integration was not significantly related to anthers’ height (GLM: t = 1.94, *p* = 0.084), throat diameter (GLM: t = 0.525, *p* = 0.612), upper lip width (GLM: t = 2.137, *p* = 0.061), and lower lip width (GLM: t = 1.614, *p* = 0.141). Model tests of PGLS were deposited in Appendix A.

The proboscis length of different pollinators of the study *Lonicera* species varied over a 16.8-fold range (from an average proboscis length of 2.45 mm in small carpenter bees to 41.15 mm in hawk moths; Appendix A). The composite pollinator proboscis length of the pollinator assemblage (a weighted proboscis length by considering all of the pollinators for each species, PLa) for *L. maackii* was the minimum, while *L. tragophylla* exhibited the longest value (Appendix A). Results of PGLS indicated that floral integration was positively related to PLa (PGLS: model = PL, b = 0.486 ± 0.119, t = 4.075, *p* = 0.0028; Figure 2e).

### 2.3. The Potential Pathway by Which Pollinators Integrate Floral Traits

The best-fitting model (Fisher’s C = 0.338, *p* = 0.844; AIC = 28.338; Appendix A for model test) showed that PLa positively related to the length of corolla tube (a trait with accessibility function; PGLS, b = 0.207 ± 0.083, *R*^2^ = 0.98, df = 310, *p* = 0.014; Figure 3) and stigma height (a trait with efficiency function; PGLS, b = 1.765 ± 0.301, *R*^2^ = 0.88, df = 310, *p* < 0.001; Figure 3), while stigma height was found to be significantly and positively related to upper lip length (a trait with attractiveness function; PGLS, b = 0.232 ± 0.03, *R*^2^ = 0.45, df = 310, *p* < 0.001; Figure 3). There were correlated errors between stigma height with corolla tube length (*p* < 0.001; Figure 3) and upper lip length with lower lip length (*p* < 0.001). PLa did not have any influence on other traits (Figure 3).

## 3. Discussion

In this study, the pollinator proboscis was found to influence the magnitude of floral integration in *Lonicera* plants. By PGLS, we found significant positive correlations between floral integration and pollinator proboscis length and four floral traits (corolla tube length, stigma height, upper lip length, and lower lip length). The best PSEM indicated that pollinator proboscis directly influenced the length of corolla tube (a trait that determine the accessibility to resources for pollinators) and stigma height (a trait with efficiency function), while upper lip length (a trait with attractiveness function) and lower lip length (attractiveness function) had a tight relationship with stigma height, which helped to maintain and to enhance the integration of honeysuckle flowers. The results, thus, illustrate a potential pathway by which pollinators may drive floral integration among closely related species.

### 3.1. Influence of Pollinator Proboscis Length on Floral Integration

Pollinator-mediated selection on floral integration has been reported in various flowering plants [3,4,11,15,31,32,33]. In *Lonicera*, the magnitude of floral integration varied largely among species, with the two species pollinated by hawkmoths, showing higher floral integration than those pollinated by bees (Table 1). Honeysuckle flowers pollinated by hawkmoths have much longer corolla tubes than bee-pollinated flowers. The long-tubed flowers may filter pollinators and form a more specialized pollination system than short-tubed flowers [34]. Several studies also indicate that plants with specialized pollination could lead to consistent selection on relevant floral traits and consequently, enhancing floral integration [12,35]. Based on investigations in contrasting populations of *Lonicera implexa*, Lázaro and Santamaría [11] showed that pollinator proboscis length could influence floral integration. Such evidence was also found in *Ruellia humilis* [31] and *Narcissus papyraceus* [16]. At a macroevolutionary scale, our results indicate that floral integration was positively correlated with the length of pollinator proboscis, confirming that pollinators could be a selection force that amplify floral integration of *Lonicera*.

### 3.2. The Potential Pathway by Which Pollinators Integrate Floral Traits

Although corolla tube length has been suggested to be evolutionarily correlated with pollinator proboscis [18,20,36,37,38,39], it is hard to disentangle which trait (corolla tube length vs. stigma height) is the main selection target of pollinators in *Lonicera*, according to our current data. However, results also show that corolla tube length has a tight relationship with stigma height, suggesting it might also be a result of genetic/developmental correlation. Therefore, pollinators may exert strong selection pressures on a target floral trait, and the genetic/developmental correlations among floral traits may trigger the enhancement of floral integration [9].

Pollinators, foraging on the most fitting flowers, obtained the highest profit, and, meanwhile, they optimized the stigmatic pollen deposition on the flowers [40,41]. Floral traits, with an attractiveness function, may increase the probability of a flower to be visited, while those with accessibility function may influence pollinators’ capability to access the resources of the flowers. Further, floral traits related to the pollination efficiency function may influence the pollen deposition on stigmas. A change in any of these floral traits may have an effect on the pollination success of the flower as a whole. For example, as the length of the corolla tube was selected by pollinators with different proboscis lengths, the position of stigma might, in flowers, also influence pollination success as pollinators change [33]. In this study, we found that both corolla tube length and stigma height were directly influenced by pollinators’ proboscis length, increasing trait-matching between pollinators and flowers, which helps to enhance pollination success across species with different pollinator assemblages.

In addition, a honeysuckle flower with large stigma height always tends to have a long (upper and lower) corolla lip. This may help to enhance the attractiveness of the flower to its pollinators [11], but it may also be related to the position of the pollinator on the landing platform [42], which, in turn, may be influenced by how long the corolla tube and stigma are relative to the pollinator’s proboscis length. The finding indicates that pollinator-mediated selection may also play an important role on the correlation pattern among floral traits [43].

## 4. Materials and Methods

### 4.1. Study Species and Sites

Field investigations were conducted at two sites: Taibai County, Shanxi Province (TB-SX) and Shennongjia Nature Reserve, Hubei Province (SNJ-HB) (see Appendix A). The sites were sampled in 2016 and 2017. In these sites, some populations included two species (Appendix A).

### 4.2. Measurements of Floral Traits and Estimation of Floral Integration

For each of the 11 *Lonicera* species, we conducted measurements of floral traits on one natural population (see Appendix A for population sizes), using a digital calliper with 0.01 mm precision (Mitutoyo, Kawasaki, Japan). Eight floral morphological traits were quantified for each of the 11 *Lonicera* species in the natural populations (each population included more than 100 individuals, Appendix A), namely, corolla tube length (the distance between the base of the corolla and the top of floral receptacle), throat diameter (the diameter of the corolla tube opening), length of upper corolla lip (the distance between the base and the top), length of lower corolla lip (the distance from the base to the top), width of upper corolla lip (the largest distance from one side to the other), width of lower corolla lip (the largest distance between one side and the other), and the height of the stigma and anthers (the distance between the base of the corolla tube and the tip of the stigma or anthers, respectively). We measured at least 30 fully opened flowers (one flower/per plant) in the population of each species. The flowers were sampled at least 5 m away from each other to make sure they were from different individuals. Specimens of plants were deposited in the Herbarium of Wuhan University (WH).

We calculated floral integration for each studied *Lonicera* species based on the eight measured floral traits. Floral integration was calculated using the index of phenotypic integration (INT), which was defined as the variance of the eigenvalues (*λi*) of a correlation matrix through a principal component analysis (PCA) of the correlation matrix [44,45]. High variance among eigenvalues means that most traits are correlated and, thus, the first principal component (PC) accounts for most of the variation (high phenotypic integration). By contrast, low variance among eigenvalues indicates that the variation within the matrix is evenly distributed among all PCs (low phenotypic integration). PCAs were conducted through the ‘PHENIX’ package [46] in *R* to calculate INT for each species. INT values were corrected as INT = (Var (*λi*) − (number of traits − 1)/number of individuals per species) and expressed as a percentage of the possible maximum integration value. The possible maximum integration value equals the number of floral traits in the correlation matrix. The significance of differences between means and confidence intervals (CI) were also scored by bootstraping (*n* = 5000 permutations in each test).

### 4.3. Pollinator Observation and Measurement of Pollinators’ Proboscis Length

In a previous pilot study, we recorded five insect guilds that visited the flowers of the 11 *Lonicera* species, namely, bumblebees, honeybees, leafcutter bees, small carpenter bees, and hawkmoths (Appendix A). In the same natural populations where we measured floral traits (Appendix A), we observed pollinator visitation in at least five inflorescences per species, using 20-min observation periods. A total of 30 observation periods were monitored for each of the species (10 h; Appendix A). We conducted pollinator diurnal observations from 1000 h to 1500 h in three to five sunny days during the peak blooming period for each species. In addition to observation during the day, for *L. japonica* and *L. tragophylla*, pollinated by moths, we also conducted nocturnal pollinator observations from 1900 h to 2400 h [27,28,47], using a red light to reduce pollinator disturbance [11]. A visitor that encountered the anther or stigma of a flower was considered a pollinator. To better indicate the contribution of different pollinator guilds to the reproduction of each plant species, we calculated the visitation rate of each pollinator guild, namely, bumblebees, honeybees, leafcutter bees, small carpenter bees, and hawkmoths, in previously marked inflorescences. The visitation rate was defined as the total number of visits by pollinators from the same guild within an observation period, divided by the total number of open flowers in the inflorescence.

In addition, we sampled pollinators and kept them in 50-mL centrifuge tubes until the measurements of their proboscis length. All the pollinator species were identified by the Institute of Zoology at the Chinese Academy of Science. The length of pollinators proboscis was measured by a digital calliper (precision 0.01 mm) after relaxing the pollinators in a humid jar for two to three days; this allowed the tissues to soften and the proboscis to be pulled out using fine forceps. At least 20 individuals were measured for each pollinator guild, except for moths (Appendix A). Morphologically specialized flowers may also be visited by multiple pollinators, rather than a single pollinator guild [48,49], and they, thus, may collectively exert selective pressures on floral traits [50]. By pollinator observations, we found pollinators with various proboscis lengths are also known to visit honeysuckle flowers. In this study, we developed a composite proboscis length by considering all of the pollinators for each *Lonicera* species. This composite length was calculated by the formula below:(1)$$\mathrm{PLa}=\sum_{i=1}^{n}\mathrm{PLi}\cdot \frac{\mathrm{VFi}}{\mathrm{VFn}}$$

PLa is the composite proboscis length of the pollinator assemblage, n is the total number of pollinator guilds in a pollinator assemblage for each plant species, PLi is the proboscis length of the ith pollinator guild (the average proboscis length of all pollinators within the guild), VFi is the visitation rate of the ith pollinator guild, and VFn is the visitation rate of the pollinator assemblage.

### 4.4. Statistical Analyses

#### 4.4.1. The Divergence of Floral Traits in 11 *Lonicera* Species

We used PCA to detect whether species differed among the 11 species in the ‘PCAtools’ package [51] of *R*. We performed a Tukey post hoc test by *R* package ‘multcomp’ [52] to determine whether species were statistically different among mean values for each trait after confirming that a difference between means in 11 species was supported by a one-way analysis of variance (ANOVA).

#### 4.4.2. The Relationship of Floral Integration with Pollinator Proboscis Length and Floral Traits

We used PGLS to test whether floral integration was related to the floral traits and the composite pollinator proboscis length (PLa). PGLS generalizes the independent contrasts approach and can be used to incorporate a variety of models of evolutionary change [53,54]. We fitted the following evolutionary models: (1) Brownian motion (BM), whose traits evolve according to random drift; (2) Pagel’s lambda (PL), which involves the rate of trait evolution that is optimized from the data; and (3) Ornstein–Uhlenbeck (OU), whose traits evolve towards an optimum (Appendix A). Then, we compared model fit using AIC [55] and used the model with the best fit to estimate the relationship. In these analyses, we used the index of phenotypic integration (INT) as a response variable, and we used the composite proboscis length of the pollinator assemblage (PLa) and floral traits as predictor variables in separate models, containing one predictor variable (to avoid overparametrization). Based on the Bayesian tree of 11 *Lonicera* species [56], PGLS analyses were conducted in the R package ‘nlme’ [57].

#### 4.4.3. The Potential Pathway by Which Pollinators Integrate Floral Traits

To disentangle the potential pathway by which the proboscis length of pollinators integrates floral traits, we applied phylogenetic structural equation modeling (PSEMs) by using the *R* package ‘piecewiseSEM’ [58]. The PSEMs comprised PGLS, using the *gls* function with the best evolutionary model in the package ‘nlme’, with Pagel’s algorithm [59], to account for evolutionary dependence among species. By this method, we built different alternative models that combined PLa and the four floral traits significantly related to INT of the 11 *Lonicera* species and then compared them to find the best-fitting one (Appendix A), which could help us to determine the key direct or indirect effects of PLa on variation and covariation in phenotypic traits. The four floral traits were classified into three groups according to functionality: accessibility (corolla tube length), efficiency (stigma height), and attractiveness (upper corolla lip length and lower corolla lip length), following previous studies [11,12]. In each model, PLa was thought to be a predictor variable that is directly or indirectly related to the floral traits from different functional modules. We created a total of 20 PSEMs, with every possible combination of pathways, due to all three conditions, wherein PLa was related to one, two, and three of the functional modules, respectively (Appendix A). Floral traits would be correlated if they are constrained by the same genetic and/or developmental basis (e.g., anther height and stigma height) [36]. Because we could not presume the relationships between floral traits within attractiveness function (upper corolla lip length and lower corolla lip length) to be causal, they were defined as being correlated errors [58]. Shipley’s test of d-separation [60,61] was used to assess the overall fit of each model, which assesses whether the model would be improved by the inclusion of identified missing paths. The d-separation test generates a Fisher’s C (Fisher, 1925) test statistic, which can be used to assess overall fit of the PSEM and to calculate AIC for model selection [60,61]. The best model was defined as the model with the lowest AIC from those with *p* values greater than 0.05, derived from Fisher’s C statistic test and with tests of d-separation, showing no statistically significant missing paths.

In this study, all data were summarized as the means ± standard errors, and all statistical tools were run in R with version 3.4.3 [62]. Except for the ANOVAs for which we applied a Bonferroni correction, the significance was considered to occur at a level of 0.05.

## 5. Conclusions

Our study illustrated a potential pathway by which pollinators integrate floral traits. Pollinator proboscis length directly influenced corolla tube length (accessibility function) and stigma height (efficiency function). In turn, covariation of stigma height with other floral traits enhances floral integration. The direct and indirect effects on floral traits might amplify the floral integration in response to a precise flower–pollinator fit. However, phylogenetic studies on the relationship between floral divergency and the pollination system, by incorporating more species, are necessary to improve our understanding of the evolution of floral integration in honeysuckle flowers.

## Figures and Tables

**Figure 1 plants-12-01629-f001:**
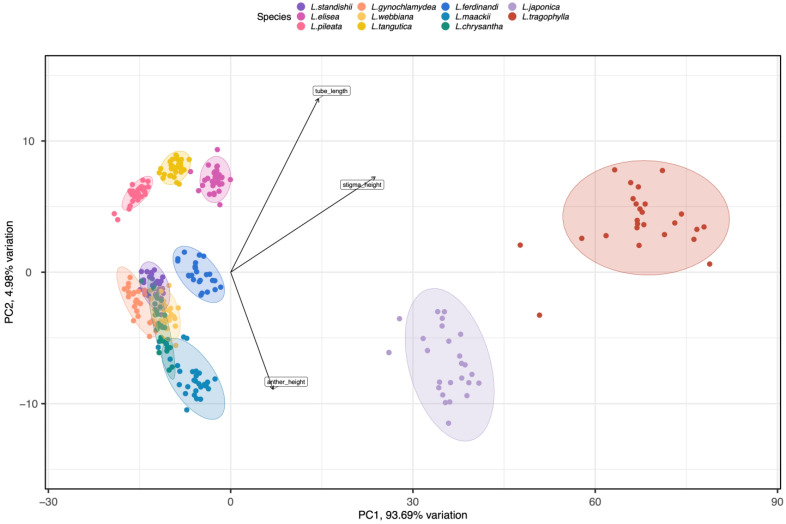
PCA analysis of eight floral traits (corolla tube length, throat diameter, anther height, stigma height, upper corolla lip length/width, and lower corolla lip length/width) among the 11 *Lonicera* species. Most variance among plant species in phenotypic traits was explained by the first (PC1) and second (PC2) principal components (93.69% and 4.99%, respectively). Tube length, stigma height, and anther height had the highest scores regarding PC1 (see arrows). Different colors represent the different study species. Ellipses include 96.13% of samples from each species.

**Figure 2 plants-12-01629-f002:**
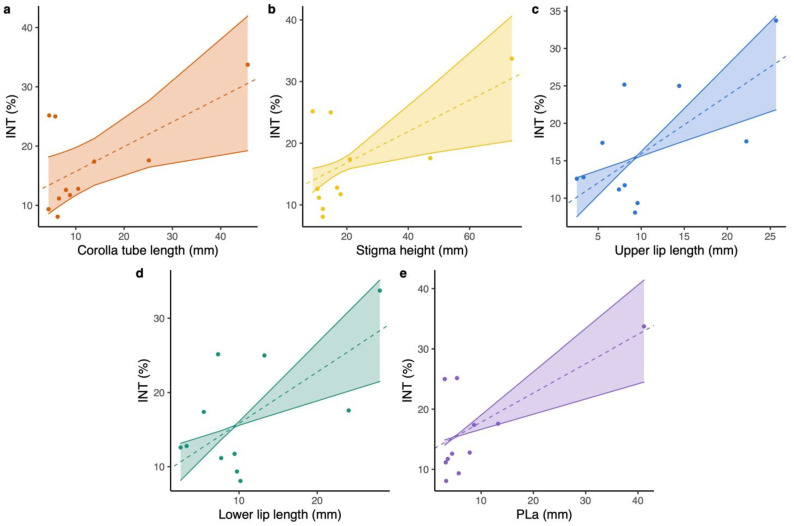
PGLS results on the correlation between: (**a**) corolla tube length and floral integration index (INT), (**b**) stigma height and INT, (**c**) upper lip length and INT, (**d**) lower lip length and INT, and (**e**) the composite pollinator proboscis length of the pollinator assemblage (PLa) and INT for the studied *Lonicera* flowers.

**Figure 3 plants-12-01629-f003:**
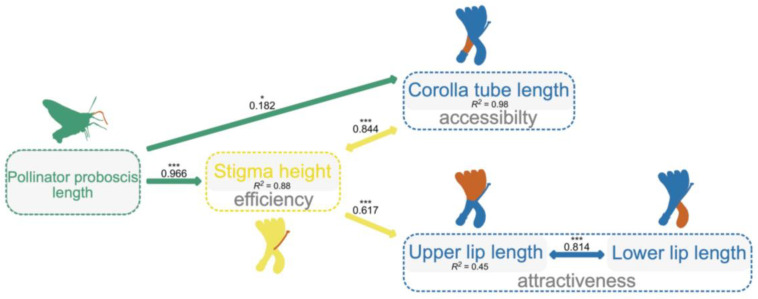
The best phylogenetic structural equation model (PSEM) shows how the pollinator proboscis length integrates floral traits for the studied *Lonicera* species. Paths between variables (highlighted in orange) included in the best-fitting model are shown. The solid arrows indicate a positive effect of a variable on another. The arrows with double ends indicate correlated errors. Standardized path coefficient was given on each arrow. *R*^2^ for each model is given below the boxes of variables. ***: *p* < 0.001, *: *p* < 0.05.

**Table 1 plants-12-01629-t001:** Floral traits (corolla tube length, throat diameter, anther height, stigma height upper corolla lip length/width, and lower corolla lip length/width), phenotypic integration index (INT), and the composite pollinator proboscis length (PLa) for the 11 *Lonicera* species. All the traits were measured in one natural study population of each species. Different letters indicate significant differences among species (Tukey post hoc tests).

Species	Corolla Tube Length (mm, N = 30)	Throat Diameter (mm, N = 30)	Anther Height (mm, N = 30)	Stigma Height (mm, N = 30)	Upper Corolla Lip Length (mm, N = 30)	Upper Corolla Lip Width (mm, N = 30)	Lower Corolla Lip Length (mm, N = 30)	Lower Corolla Lip Width (mm, N = 30)	Phenotypic Integration Index (%)	PLa (mm)
*L. chrysantha*	4.36 ± 0.53 ^h^	3.14 ± 0.044 ^bc^	7.75 ± 0.237 ^bc^	12.2 ± 0.115 ^f^	9.57 ± 0.097 ^d^	3.12 ± 0.043 ^cd^	9.73 ± 0.115 ^d^	3.05 ± 0.04 ^ef^	9.346	5.711
*L. elisae*	13.83 ± 0.15 ^c^	1.86 ± 0.046 ^g^	6.98 ± 0.217 ^c^	20.98 ± 0.267 ^c^	5.49 ± 0.091 ^f^	4.66 ± 0.073 ^b^	5.49 ± 0.091 ^f^	4.66 ± 0.073 ^a^	17.387	8.64
*L. ferdinandi*	8.8 ± 0.47 ^e^	2.94 ± 0.018 ^cd^	8.77 ± 0.220 ^b^	17.99 ± 0.278 ^d^	8.08 ± 0.069 ^e^	2.63 ± 0.030 ^e^	9.42 ± 0.082 ^d^	2.51 ± 0.028 ^g^	11.727	3.619
*L. gynochlamydea*	4.5 ± 0.29 ^gh^	2.32 ± 0.02 ^ef^	7.07 ± 0.312 ^c^	8.87 ± 0.157 ^g^	8.04 ± 0.027 ^e^	2.69 ± 0.117 ^de^	7.31 ± 0.032 ^e^	2.79 ± 0.11 ^fg^	25.164	4.457
*L. japonica*	25.12 ± 0.65 ^b^	3.51 ± 0.057 ^b^	22.12 ± 0.245 ^a^	47.18 ± 0.462 ^b^	22.21 ± 0.112 ^b^	4.27 ± 0.049 ^b^	24.02 ± 0.132 ^b^	3.91 ± 0.046 ^bc^	17.583	13.253
*L. maackii*	5.76 ± 0.48 ^fg^	2.59 ± 0.031 ^de^	7.98 ± 0.2284 ^bc^	14.76 ± 0.347 ^e^	14.41 ± 0.083 ^c^	3.44 ± 0.037 ^c^	13.24 ± 0.099 ^c^	3.49 ± 0.034 ^cd^	25.002	3.045
*L. pileata*	7.97 ± 0.25 ^e^	2.25 ± 0.033 ^eg^	5.44 ± 0.23 ^de^	10.45 ± 0.259 ^fg^	2.52 ± 0.131 ^g^	2.55 ± 0.0548 ^e^	2.52 ± 0.157 ^g^	2.55 ± 0.051 ^g^	12.597	5.392
*L. standishii*	6.52 ± 0.1 ^f^	2.1 ± 0.058 ^fg^	7.74 ± 0.299 ^bc^	10.93 ± 0.159 ^f^	7.41 ± 0.15 ^e^	2.57 ± 0.069 ^e^	7.71 ± 0.175 ^e^	2.66 ± 0.066 ^fg^	11.16	3.239
*L. tangutica*	10.5 ± 0.64 ^d^	4.46 ± 0.055 ^a^	4.51 ± 0.212 ^e^	16.8 ± 0.077 ^d^	3.29 ± 0.055 ^g^	3.26 ± 0.024 ^c^	3.29 ± 0.066 ^g^	3.26 ± 0.023 ^de^	12.792	7.817
*L. tragophylla*	45.59 ± 0.71 ^a^	4.34 ± 0.069 ^a^	21.64 ± 0.253 ^a^	73.76 ± 1.218 ^a^	25.66 ± 0.126 ^a^	5.48 ± 0.055 ^a^	28 ± 0.149 ^a^	5.1 ± 0.052 ^a^	33.724	41.15
*L. webbiana*	6.25 ± 0.31 ^f^	3.33 ± 0.073 ^bc^	6.84 ± 0.204 ^cd^	12.15 ± 0.18 ^f^	9.28 ± 0.041 ^d^	3.51 ± 0.179 ^c^	10.17 ± 0.049 ^d^	4.21 ± 0.168 ^b^	8.077	3.311

## Data Availability

Data are contained within the article or Appendix A.

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
