# Peer review of "Pollinator Proboscis Length Plays a Key Role in Floral Integration of Honeysuckle Flowers (Lonicera spp.)"

_plants, 2023, doi:10.3390/plants12081629_

Round 1
Reviewer 1 Report
Reviewer's comments and suggestions are attached in pdf of the paper and in separate file in zip

Author Response
Response to Reviewer 1 (Bold and Italics)
- General opinion about the paper. The topic is interesting focused on the flower structure evolution in relation to the insect visitors proboscis length based on 11 Lonicera species research. Regarding that the genus Lonicera is the largest genus in the Caprifoliaceae, rich in species (over 180) largely distributed in temperate to subtropical regions of the northern hemisphere: Europe, Asia and North America, research on 11 species are not enough to take so general conclusion about the floral structure evolution under the pressure of proboscis length of insect pollinators. The genus is subdivided on subgenera and sections. Authors should include such information to studied species. Moreover, there is no link between the results and the genus phylogeny based on molecular markers (e.g., Theis et al. 2008, Nakaji et al. 2015). It would be interesting to show a location the studied species on the tree, what is the evolutionary trend.
The novelty of the studies should be clearly indicated because several papers on Lonicera were already published on this topic, using similar methods and statistics e.g. Lázaro, A.; Santamaría, L. Flower-visitor selection….(in References nr 11), Lázaro, A.; Vignolo C.; Santamaría, L. Long corollas as..(In References nr 29).
Thank you very much for the comments.
First of all, we agree with you that 11 species from a genus with relatively large numbers of species are not enough to present an evolutionary story. That is why we did not try to illustrate the evolutionary trends of the floral traits involved in this study although the 11 species were distributed in eight sections of the genus according to the pervious study (Smith, 2009). However, in this study, we focused on testing the potential of pollinators driving floral integration in Lonicera. In this circumstance, our detailed data on field investigations across the 11 phylogenetically related species provided a well framework to study the relationship between variation in floral integration and pollinator composition.
Although there are some studies on floral integration of Lonicera species, a comparatively approach involving multiple species with different pollination system is still necessary to better understand the substantial floral variation across species because the studies you mentioned (references 11 and 29) focused on intraspecific variation among populations. Therefore, we thought the novelty of our study is whether and how pollinators influencing floral integration.
Detailed comments to Authors:
- Title: Please, add latin name of the genus to the title
Thank you very much for the comment. The Latin name has been added into the title.
- Abstract: lines 18-19: Authors wrote: “potential pathway through which pollinators drive floral integration remains unexplored” see papers of Beattie on Viola and Grant on Phlox and also on Aquilegia (references are listed)
Thank you very much for the comment. We now weakened the expression and the sentence was changed as “potential pathway through which pollinators drive floral integration needs further investigations”. (Line 19)
- Introduction is too general. Authors should:
- a) information about the genus Lonicera should be included into the Introduction (species number, distribution, the genus taxonomy: subgenera, sections). Isolating mechanisms of species pre- or post zygotic? (move information from Mat&Meth to Introduction); hybridization between species is phenomenon frequent in populations of studied species? In population from which material was sampled in the present paper, several or only one species of Lonicera occur? Mixed or monospecies populations? Important information in relation to pollination vectors
Thank you very much for the suggests. Information of Lonicera genus was removed into Introduction accordingly. There are no data ( from references or Flora) about isolating mechanisms of Lonicera species. Based on our filed experiments (nonpublished data), pollinators may play a key role on reproductive isolation. At the same time, we did not find hybrid species in our study sites. But there are few cultivated hybrid spices (e.g., Lonicera×heckrottii). We also added populations (Mixed or monospecies) information into the Methods. (Lines 225-226)
- b) briefly describe the structure of the Lonicera flowers and add drawings or photos of all investigated species, especially stigma and corolla. Authors should move lines 2010-2018 from Materials and Methods to the Introduction. It is really difficult to jump to the Mat&Meth to find information about flower structure
Thank you very much for the suggests. We added pictures (as Figure S1) of 11 species into Introduction. (Line 88)
- c) add important information about pollination system - self or exclusively cross-pollination or mixed system. As above, move information from Mat&Meth to the Introduction
Thank you very much for the suggests. We have removed them into Introduction. (Lines 88-89)
- d) add information about type of flowers; stenotribic or nototribic (depending on pollinator position head up or head down, see Beattie papers); add information about flower type of studied species: pollen or nectar flowers?
Thank you very much for the suggests. Lonicera species studied in this study provided both pollen and nectar for pollinators and we now added the information. (Line 90)
We did not add information on stenotribic or nototribic pollination for the flowers in our study because a single species could be visited by several different pollinator guilds and those displayed substantial variation in visitation behavior.
- e) add the influence of environmental factors on floral morphology, especially in regions with reduced number of insect visitors
Thank you very much for the comments. We agree with you that environmental factors highly affect pollination intensity and indirectly influence pollinator-mediated flower evolution. However, the 11 Lonicera species were observed in the two sites with similar environmental conditions. Moreover, pollinator composition was substantially varied among them. We therefore infer that the variation in pollinators among species might not be influenced by environmental factors.
- f) add that flower color is important trait playing a role in speciation; flower color is responsible for pollinator-mediated reproductive isolation in areas where the two species co-occur eg.: Mimulus
Thank you very much for the comments. Details of floral color were added into Introduction of this revised version. (Lines 75-76)
- g) provide information about genetic background of flower structure diversification. The genetic and genomic resources has provided insight into how the traits arose and diversified
Thank you very much for the comment. We agree with that genetic and genomic resources can provide insight into flower diversification. However, except for phylogenetic studies on Lonicera, we did not find more genetic and genomic resources for this taxon. In addition, one of the reasons for our selection of the 11 plant species is the substantial variation in pollinator composition. Although flower diversification could be influenced by genetic differentiation, our goal of this present study is to test whether and how pollinators influence flower diversification.
- h) mention in Introduction important papers of Beattie on flower evolution in Viola, Verne Grant in Phlox and also in Aquilegia (references are listed in comments to Authors) and mention about Darwin orchid and moth – correlation between long orchid spur and long proboscis of moth
Thank you very much for the comments. These references were cited in Introduction accordingly.
Results:
- 5. Subtitles: 2.1.1., 2.1.2, 2.1.3 are description of results
2.2 is separated as Figures, Tables and Schemes; do not separate this subtitle because Figures, Tables and Schemes are cited in the previous subtitles of Results
2.3. is description of method not results
Thank you very much for the comments. There were some subtitles of figures due to the manuscript template from the journal’s instructions for authors (more detail see https://www.mdpi.com/journal/plants/instructions).
To enhance the reading experience, we removed 2.3 to Methods (Lines 287-293)
- Material and Methods:
- a) In Table S1 Authors should add information: to each subgenus and section belong studied species
Thank you very much for the suggestion. The information was added accordingly.
- b) In Table S1 Population size: an area of the population should be given; the term population size is misleading as it means the number of species but it is important to know the area of the population?
Thank you very much for the suggestion. We agree with you that population area should be also provided. However, populations of some studied species located in habitats of mountain slope; we can roughly count the number of individual plants but we failed to evaluate the population area in the field.
- c) In Table S1 the GPS coordinates are provided. There is lack of information about height above sea level. Suggestion is to add a map with location of studied populations
Thank you very much for the suggestion. Information of sea level was added accordingly. We did not add a map because our study was conducted in only two sites, each site with several sympatric species.
- d) In Table S1 add information from which area particular species were analyzed; based on coordinates it is hard to deduce
Thank you very much for the comment. We conducted field investigations in two sites; each stie contained several sympatric Lonicera species. That meant different species shared a similar habitat.
- e) Information about studied populations should be provided in Materials and Methods
Thank you very much for the comment. The information was added accordingly. (Lines 225-226)
- f) 30 flowers per population, please write more precisely: one flower/per plant ? meaning 30 plants per population but one flower per plant??
Thank you very much for the comment. We measured 30 flowers each from different individual plants. The sentence has been revised accordingly. (Line 239)
Discussion
References suggested to Authors:
- Beattie AJ. 1971. Pollination Mechanisms in Viola. New Phytol. 70(2): 239-444.
- Beccaloni, G. W. 2017. Wallace's moth and Darwin's orchid. pdf attached
- Bharti Sharma, Levi Yant, Scott A Hodges, Elena M Kramer. 2014. Understanding the development and evolution of novel floral form in Aquilegia. Current Opinion in Plant Biology 17: 22-27
- Bieniasz M, Dziedzic E, Kusibab T. 2022. Evaluation of morphological traits of flowers and crossing possibility of haskap (Lonicera L) cultivars depending on their origin. Journal of Berry Research 12(3):1-15, DOI: 10.3233/JBR-211507
- Grant V., Grant KA. 1965. Flower Pollination in the Phlox Family. Columbia University Press.
- Toji T. et al. 2022. Intraspecific independent evolution of floral spur length in response to local flower visitor size in Japanese Aquilegia in different mountain regions. Ecology and Evolution 12
Thank you very much for providing these literatures. We now carefully read the literatures and cited them in revised version. We hope these citations could make the discussion more reasonable.

Reviewer 2 Report
The manuscript examined floral integration in 11 Lonicera species based on the measurements of floral traits and pathway of floral integration driven by pollinators. Generally, the manuscript was well organized, and my concerns were in the following.
1. Line 95: L. tragophylla should be presented as italic.
2. Line 115, 117, 120, 121: Figure 3 should be Fig. 3
3. Line 145: Please give the references on the formula. If the formula is developed by the authors, it would be better to present the brief introduction on the deduction of the formula. In addition, it seems to be better for the paragraph to be moved in MM section.
4. Line 162: I don’t think “integrate” is suitable here.
5. Line 227-237: It would better to provide a schematic figure to illustrate the five measured floral traits, and this would make the traits clearer than present.
Author Response
Response to Reviewer 2 (Bold and Italics)
- The manuscript examined floral integration in 11 Lonicera species based on the measurements of floral traits and pathway of floral integration driven by pollinators. Generally, the manuscript was well organized, and my concerns were in the following.
Thank you so much for the compliments on our work. We carefully revised the manuscript according to all the very helpful comments and suggestions you provided.
- Line 95: L. tragophylla should be presented as italic.
Thank you so much for the careful correction, which was changed accordingly.
- Line 115, 117, 120, 121: Figure 3 should be Fig. 3
Thank you so much for the careful corrections, which were changed accordingly.
- Line 145: Please give the references on the formula. If the formula is developed by the authors, it would be better to present the brief introduction on the deduction of the formula. In addition, it seems to be better for the paragraph to be moved in MM section.
Thank you so much for the comment. We have removed them into the Methods with a brief introduction of the formula.
- Line 162: I don’t think “integrate” is suitable here.
Thank you so much for the comment. This sentence was revised as “…which pollinators may drive floral integration among closely related species.”. (Lines 174-175)
- Line 227-237: It would better to provide a schematic figure to illustrate the five measured floral traits, and this would make the traits clearer than present.
Thank you very much for this suggestion. The eight floral traits are common floral traits that measured by some other studies (e.g., Lázaro and Santamaría, 2016). In this revised manuscript, we added a figure to show the flowers of the 11 Lonicera species, which may help readers to understand the method for measurement of the floral traits.

Round 2
Reviewer 1 Report
Reviewer’s comments to revised manuscript
Pollinator proboscis length plays a key role in floral integration of honeysuckle flowers (Lonicera ssp.) by Gan-Ju Xiang et al.
Authors improved manuscript according to reviewer’s comments and suggestions. Some minor corrections are needed: Materials & Methods. Suggestion is to add abbreviation of the site name and provided this abbreviation to the Table S1 in the coordinates column. Please, see attached comments and suggestion.
Author Response
Thank you very much for the suggestion. Abbreviations of the site names were added into Materials & Methods and Table S1.
